# Evidence of Delta Phase of Fe in MBE-Grown Thin Epitaxial Films on GaAs

**Ramasis Goswami * and Syed B. Qadri**

Materials Science and Technology Division, Naval Research Laboratory, Washington, DC 20375, USA; syed.qadri@nrl.navy.mil
* Correspondence: ramasis.goswami@nrl.navy.mil

**Abstract:** Fe/GaAs is an important system for the study of spin injection behavior that can vary with the nature and interfaces of Fe films. Here, we investigate the effect of interfacial strain on the microstructure, interfaces and phase-formation behavior in epitaxially grown Fe films. To vary the strain, we have characterized Fe films of various thicknesses ranging from 10 to 1000 nm which were grown using molecular beam epitaxy on GaAs (011) and AlGaAs (001) substrates. High resolution X-ray diffraction studies revealed that films with higher thicknesses exhibited an equilibrium α-Fe phase, while the films with less than 10 nm thicknesses indicated the presence of δ-Fe. Transmission electron microscopy revealed the interface for 10-nm-thick films had strain lobes with no interfacial phase formation for films deposited at room temperature. At a higher deposition temperature of 175 °C, similar strain lobes were observed for a 10-nm-thick film. Extended annealing at 200 °C transformed the metastable δ-Fe phase to an equilibrium α-Fe. However, at higher temperature, the interface contained an intermixing layer of (FeAl)GaAs. We demonstrate that the interfacial strain plays a major role in stabilizing the metastable δ-Fe on GaAs.

**Keywords:** TEM; microstructure; thin film; defects; interfaces; magnetic semiconductors





## 1. Introduction

Fe/GaAs is an interesting system to study spin injection behavior [1–6] because it exhibits high spin polarization. Considerable work has been done and it has been theoretically suggested that an abrupt Fe/GaAs interface with no intermixing and no interfacial reaction products would be required for efficient spin injection [7–9]. Recently an enhancement of spin injection efficiency has been reported by the post-growth annealing of Fe/GaAs films [10]. Although considerable theoretical, as well as experimental, efforts have been made to study the spin injection behavior, few investigations have focused on the nature and structure of Fe films on GaAs as a function of thickness, as the lattice spacing of Fe on GaAs can vary with the thickness of Fe due to lattice-mismatch-induced strain. It has been observed, using ferromagnetic resonance (FMR) and SQUID studies on Fe/GaAs films, that the magnetic moment per unit volume decreases with decrease in film thickness [11–14]. The theoretical calculation also showed a decrease in the magnetic moment per unit volume with decrease in the lattice parameter of bcc-Fe[14]. It has been argued that the post-annealing modifies the atomic arrangement of the Fe/GaAs interface; different interface structures have been reported in the literature[15]. Upon post-growth annealing of Fe/GaAs at 200 °C for 1 h, the interface structure consists of a partial layer of Fe atoms between the Fe film and the GaAs [15]. At elevated growth temperature (>250 °C), a number of interfacial phases [16–18], $Fe_2As$, $Fe_3Ga$, $Fe_3Ga_{2-x}As_x$, have been reported. However, at lower growth temperatures, an extended interfacial region of Fe, Ga and As has been observed [19].

Recently, we have shown that strain plays a major role in dictating the crystal structure of AlN on a GaN/sapphire (0001) substrate. The AlN film forms a metastable zinc-blende

cubic phase at relatively low temperatures, and the strain, estimated in the observed orientation relation, is significantly lower for cubic AlN on hexagonal GaN compared to the hexagonal AlN on hexagonal GaN [20]. It would, therefore, be important to examine whether the strain at the interface affects the structure of Fe film on GaAs. We have revisited the studies of the nature and structure of Fe film samples, grown at room and higher temperatures, by high-resolution X-ray diffraction (XRD) and transmission electron microscopy (TEM) techniques. Here, we show that, as the film thickness decreases from 1000 nm to 10 nm, a metastable δ-phase of Fe is stabilized on GaAs at lower temperatures. For the film thickness of 10 nm, the observed lattice parameter of bcc phase is 2.83 Å, consistent with the δ-Fe phase. Upon annealing at 200 °C, the lattice parameter increases to 2.86 Å, which is very close to the lattice parameter of α-Fe. No other reaction layer was observed on films deposited at room temperature and at 175 °C and upon further annealing at 200 °C.

## 2. Experimental Techniques

Fe films of various thickness, 10 to 1000 nm, were selected that were grown on GaAs (110) and AlGaAs/GaAs (001) templates at room temperature and at 175 °C using molecular beam epitaxy (MBE) at a growth rate of 0.5 nm/min. The room temperature grown samples were then annealed at 200 °C for one hour. For more experimental details, see [11–14]. To study the crystal structure of Fe films on GaAs, XRD scans were performed with monochromatic $CuK_{\alpha1}$ radiation using a Rigaku 18 kW rotating anode generator and a high-resolution powder diffractometer. ω-2θ scans were made of the films to determine the interplanar d-spacings normal to the films and substrates in specular or reflection mode. A JEOL 2200-FX transmission electron microscope was used to characterize the microstructure and interfaces of the Fe/AlGaAs. For cross-sectional TEM, initially, two 1.5 mm × 2 mm samples were glued together, and then mechanically polished on diamond-coated papers to a thickness of 25–50 μm. Final thinning was carried out using a Gatan ion mill with Ar-ion with a gun voltage of 5 kV and a sputtering angle of 10°. We performed fine-probe energy dispersive spectroscopy (EDS) mapping (probe size 2 nm) in scanning TEM (STEM) mode to study the distribution and intermixing of Fe in GaAs. High-angle annular dark field (HAADF) imaging was also performed to study different layers. Fast-Fourier transforms (FFTs) and inverse-FFT (IFFT) were obtained from experimental high-resolution transmission electron microscopic (HRTEM) images using Digitalmicrograph<sup>TM</sup> software.

## 3. Results and Discussion

As the misfit of bcc-αFe on GaAs lattice parameters was relatively high, the bcc Fe became compressed in order to be commensurate with the GaAs. The high resolution XRD studies showed changes in lattice spacing as a function of Fe film thickness. Figure 1a shows the X-ray diffraction studies of the Fe film on GaAs of ~1000 nm thickness, showing the (110) lattice spacing of Fe was 2.0257 Å. For a thickness of ~200 nm (Figure 1b), the (110) lattice spacing of Fe was approximately 2.021 Å, suggesting that the lattice parameter changed from 2.865 to 2.845 Å. The change in lattice parameter was significantly higher for film thickness ≤10 nm. For 10-nm-thick film on (001) AlGaAs at room temperature (see Figure 2a), the 200 peak of Fe appeared at a 2θ = 65.815°, corresponding to a lattice spacing of 1.4173 Å, and the lattice parameter was ≈2.835 Å. A similar result was obtained for 10-nm-thick film at 175 °C (see Figure 2a). The lattice spacing of the (200) at 2θ = 65.815° was 1.4173 Å, and the lattice parameter was ≈2.835 Å.

Figure 3 shows the lattice parameter of the film as a function of Fe film thickness in the range of 10 to 1000 nm, exhibiting the lattice parameter decreasing with decrease in thickness in a non-linear fashion. The measured lattice parameter of bcc-Fe on GaAs for the 10-nm-thick film was 2.835 Å, which was consistent with the δ-Fe lattice parameter, which was smaller than that of the equilibrium alpha phase of bcc Fe (α-Fe) at room temperature. Based on the lattice parameter, the Fe-film had adopted the metastable δ-Fe phase of Fe. Such metastable phases have been observed on other materials. For example, CdSe grown

on GaAs forms a zinc blende phase instead of an equilibrium wurtzite phase under ambient conditions [12]. Thus, in this case, the 10-nm-thick Fe film on GaAs was in a metastable state in the as-deposited condition, which reduced the misfit strain to 0.5%, compared to a misfit of 1.3% for the equilibrium bcc α-Fe phase on GaAs. The bcc delta phase of Fe was an equilibrium phase at high temperature. Such a metastable δ-Fe phase could transform to the stable bcc-Fe (α-Fe) phase on annealing at relatively low temperatures. Note that the Fe transformed from bcc α-Fe to fcc γ-Fe at 910 °C, and then to bcc δ-Fe at 1394 °C, before melting at 1538 °C. The lattice parameters of the α-Fe and δ-Fe phases were 2.866 Å and 2.835 Å, respectively.

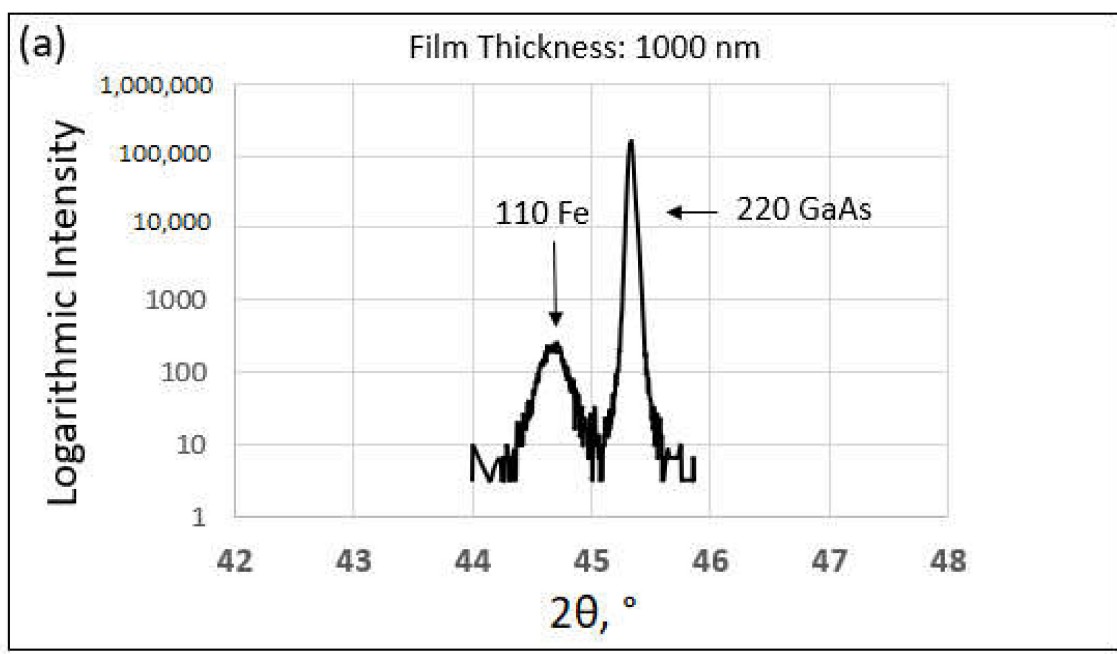

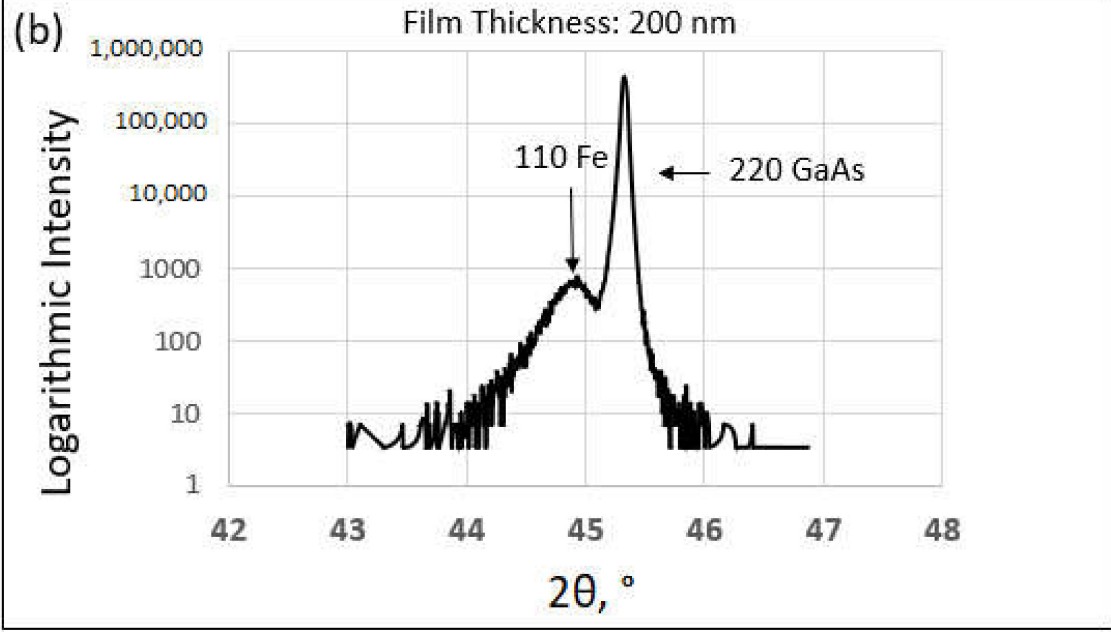

**Figure 1.** (**a**) XRD of 1000-nm-thick Fe film deposited at room temperature on (110) GaAs, showing the 110 peak of α-Fe phase corresponding to a lattice parameter of 2.865 Å. (**b**) XRD of 200-nm-thick Fe film deposited at room temperature on (110) GaAs showing a lattice parameter of 2.845 Å.

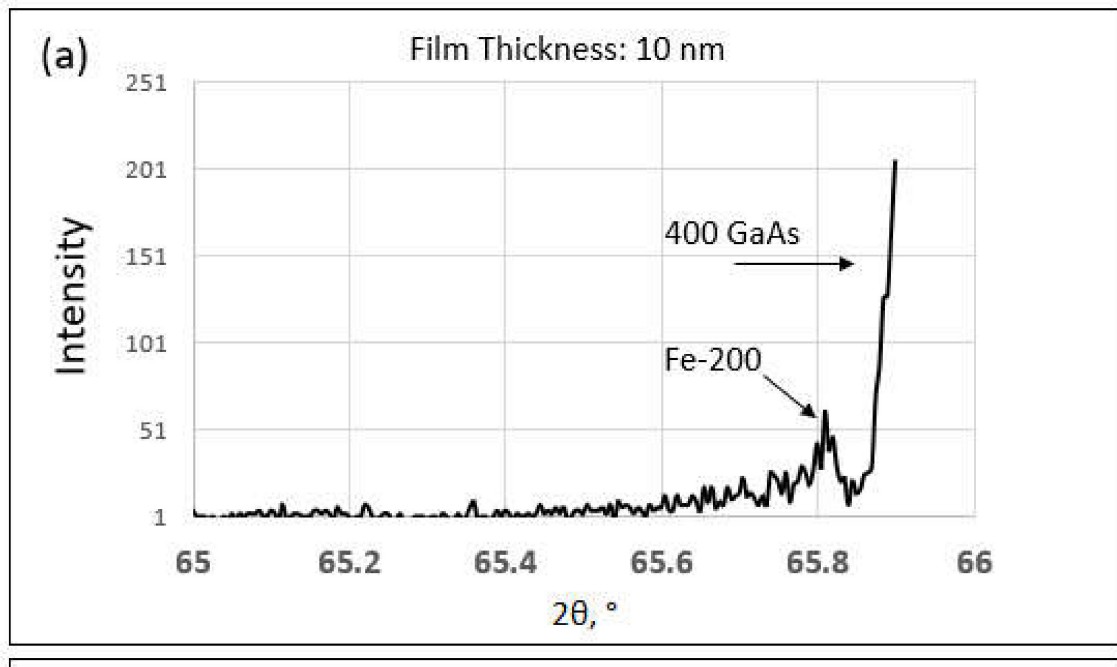

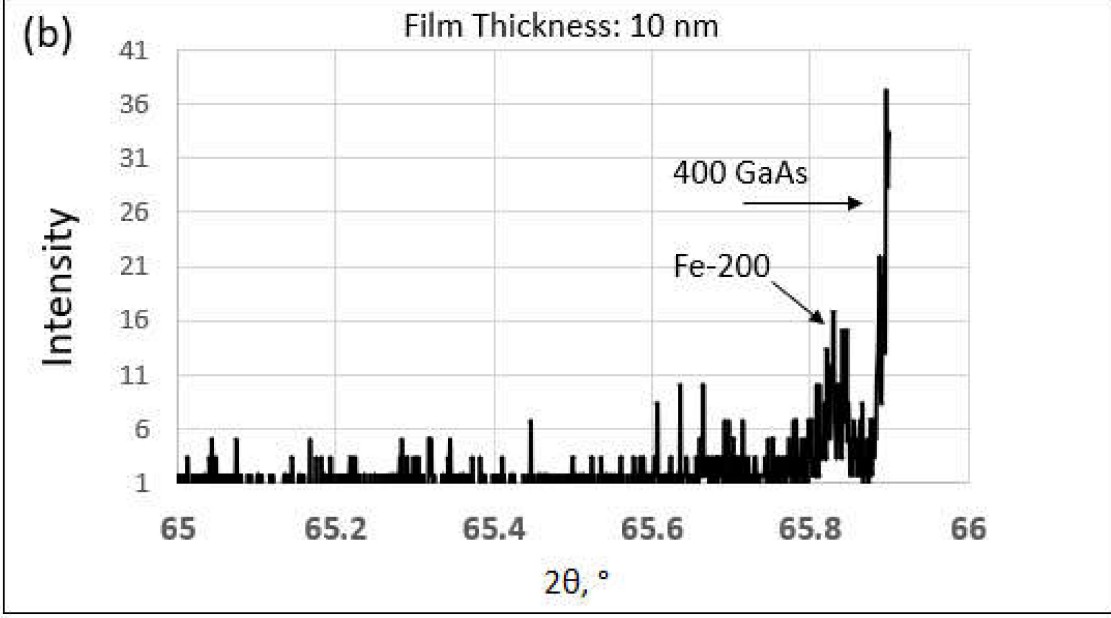

**Figure 2.** (**a**) XRD of 10-nm-thick Fe film deposited at room temperature on (001) AlGaAs, showing the 002 peak, corresponding to a lattice parameter of 2.83 Å. (**b**) XRD of 10-nm-thick Fe film deposited at 175 °C on (001) AlGaAs, showing the 002 peak, corresponding to a lattice parameter of 2.833 Å.

As the misfit of bcc δ-Fe on (001) AlGaAs was approximately 0.5% for 10 nm thin film, the Fe/GaAs interface was likely to be coherent with some amount of coherency strain. We observed that the growth of the Fe films was epitaxial on the AlGaAs. The Fe films exhibited an orientation relationship with AlGaAs that can be written as $(220)_{AlGaAs} || (110)_{Fe}$ and $[1–10]_{AlGaAs} || [1–10]_{Fe}$. Figure 4a is the bright-field image obtained very close to the two-beam diffraction condition, i.e., a transmitted beam and a strongly diffracted (004) spot. The diffraction condition is shown in Figure 4b. Under this diffracting condition, dark lobes emanating from the interface were observed. The presence of such lobes indicates that the interface was associated with the strain field, presumably because of the presence of strain at the interface. Similar strain lobes were observed after annealing the sample at 175 °C (see Figure 4c), suggesting that the interface was strained at higher deposition temperature. The diffraction condition for this image is shown in Figure 4d.

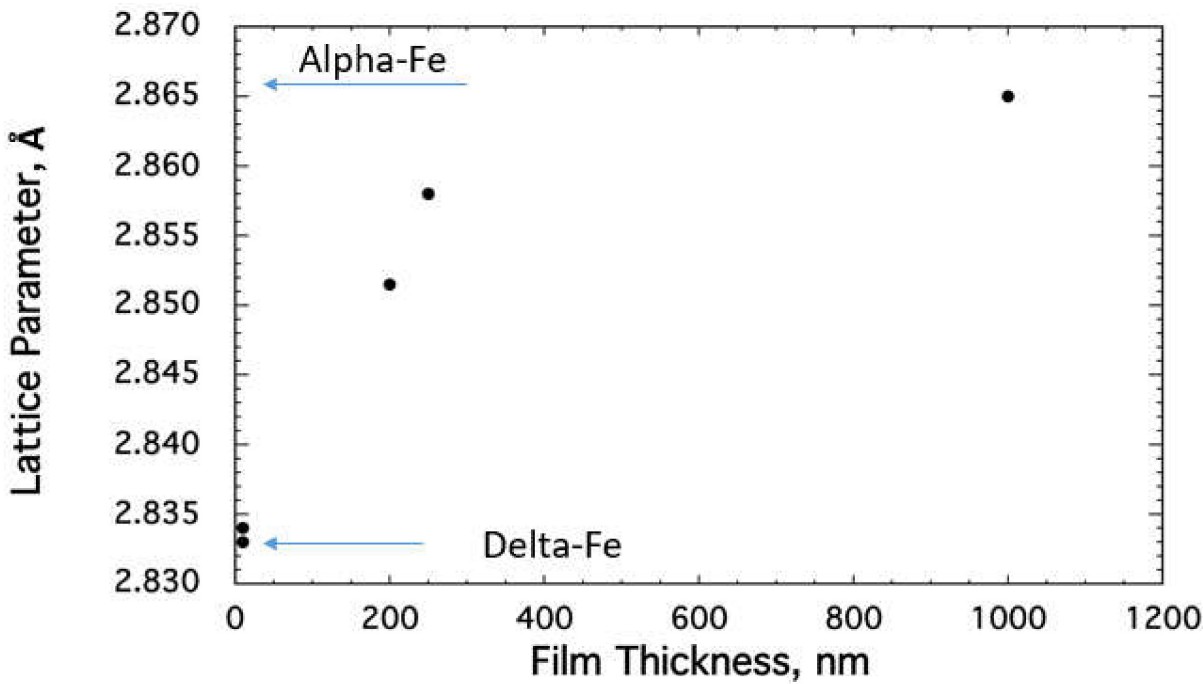

**Figure 3.** A plot showing the change in lattice parameter as a function of Fe film thickness on GaAs. The lattice parameter of $\alpha$ and $\delta$-Fe has been indicated with an arrow.

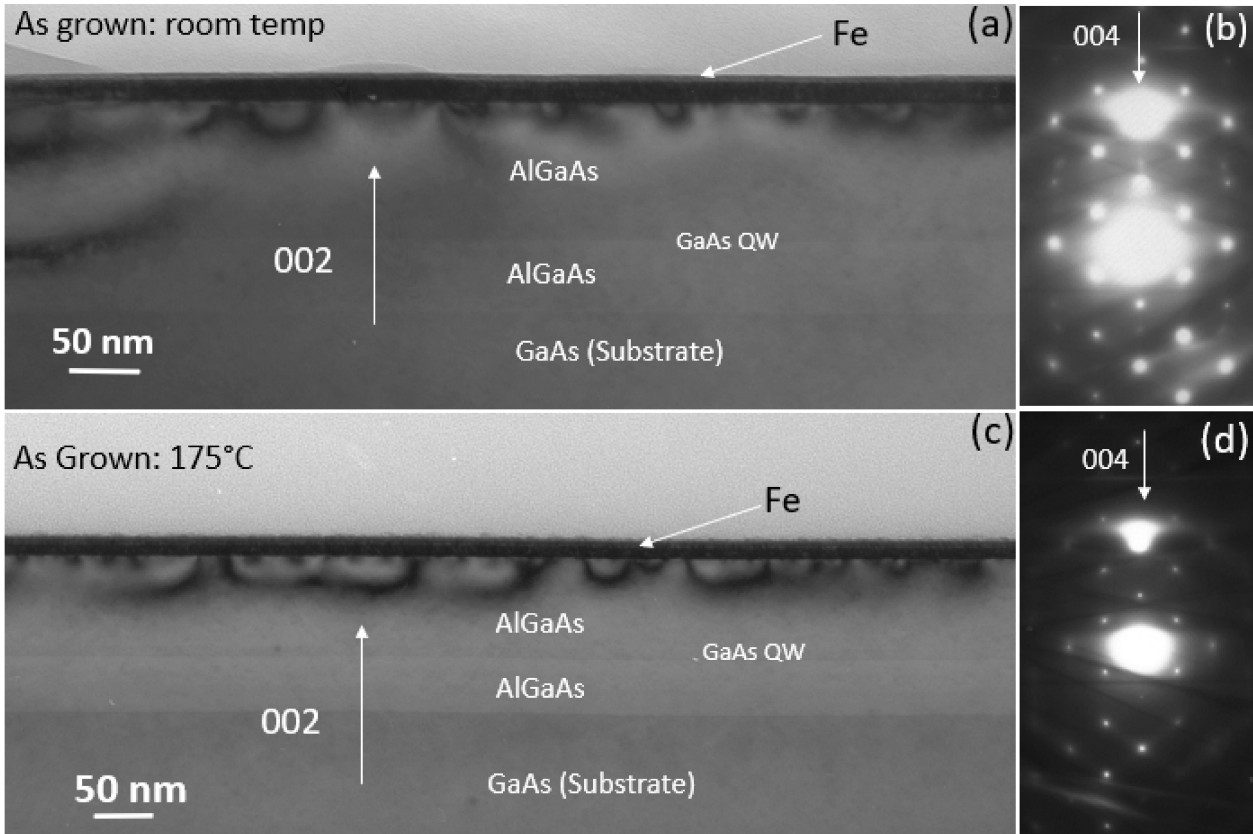

**Figure 4.** (**a**) A bright-field TEM image showing the strain lobes at the Fe/GaAs interface for the room-temperature-grown sample. (**b**) The corresponding two-beam diffraction condition. (**c**) A bright-field TEM image showing the strain lobes at the Fe/GaAs interface for the 175 °C-grown sample. (**d**) The corresponding two-beam diffraction condition.

To investigate the structure close to the interface, detailed high-resolution transmission electron microscopy was performed for the as-grown samples at room temperature and at 175 °C. Figure 5a shows the HRTEM image of the Fe and AlGaAs layers, close to the [1–10] zone of AlGaAs, for the sample deposited at room temperature. The Fe/AlGaAs interface plane was 002 of Fe and AlGaAs. The FFT obtained from the Fe and AlGaAs layers is given in Figure 5b, showing the orientation relation between Fe and AlGaAs. No reaction product was observed at the interface. The film deposited at 175 °C is given in (Figure 5c), showing the Fe/AlGaAs interface, which looked similar to that of the as-deposited film at room temperature, and no reaction product could be observed at elevated temperature. For comparison, we show here the IFFT image of the Fe/AlGaAs interface and the schematic diagram Fe/GaAs interface in the observed orientation relation, respectively, in Figure 5d,e. In addition, a number of dislocations were observed in the Fe film. Figure 6 is the IFFT-HRTEM image obtained using the reflections of $220_{AlGaAs}$ and $110_{Fe}$ of the room-temperature-deposited film showing a number of dislocations. Such dislocations in the film can form due to small misorientations that might occur due to the 3D (island type) of growth of Fe film on AlGaAs.

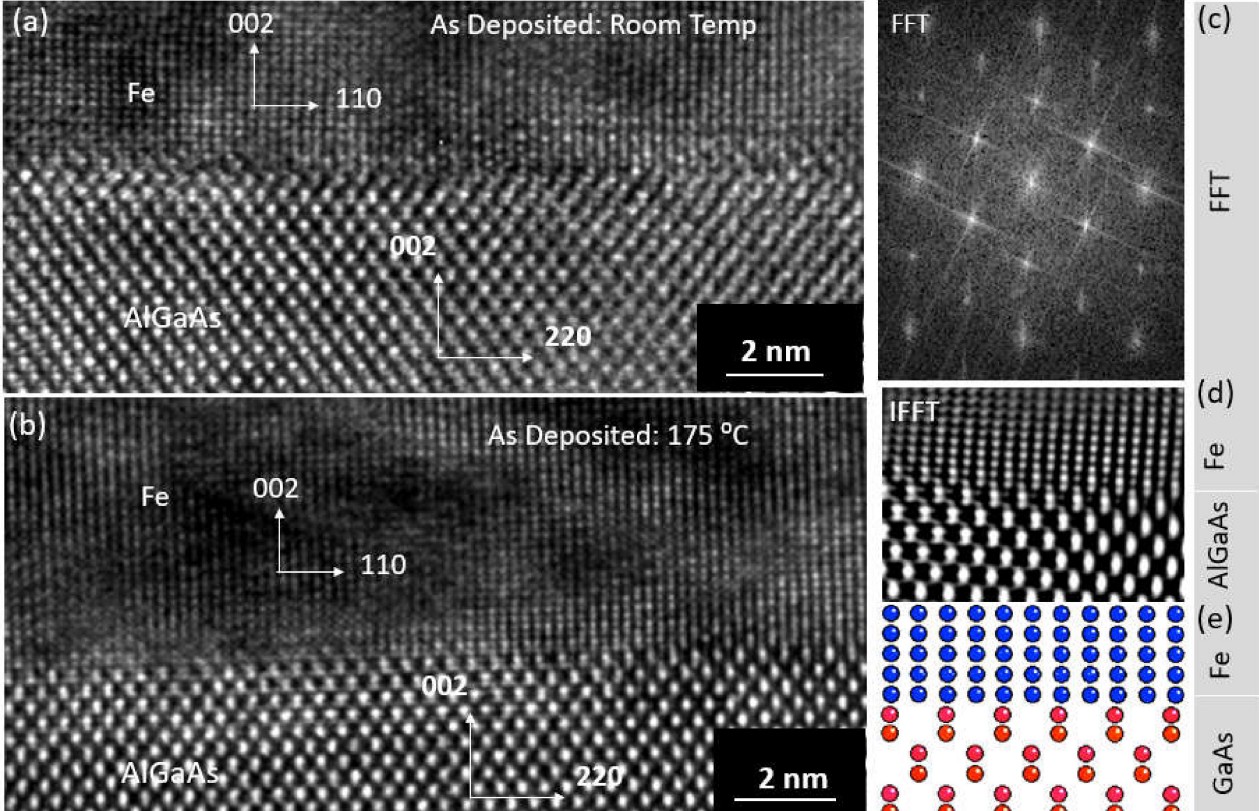

**Figure 5.** (**a**) A HRTEM image of the film grown at room temperature showing the structure of Fe and AlGaAs layers, and the interface close to the [011] zone of AlGaAs; (**b**) The corresponding FFT from both layers is shown as a right inset; (**c**) An HRTEM image of the film grown at 175 °C, showing the structure of Fe and GaAs layers, and the interface close to the [011] zone of AlGaAs; (**d**) The IFFT image; (**e**) A schematic diagram of the Fe/GaAs interface structure in the observed orientation relation.

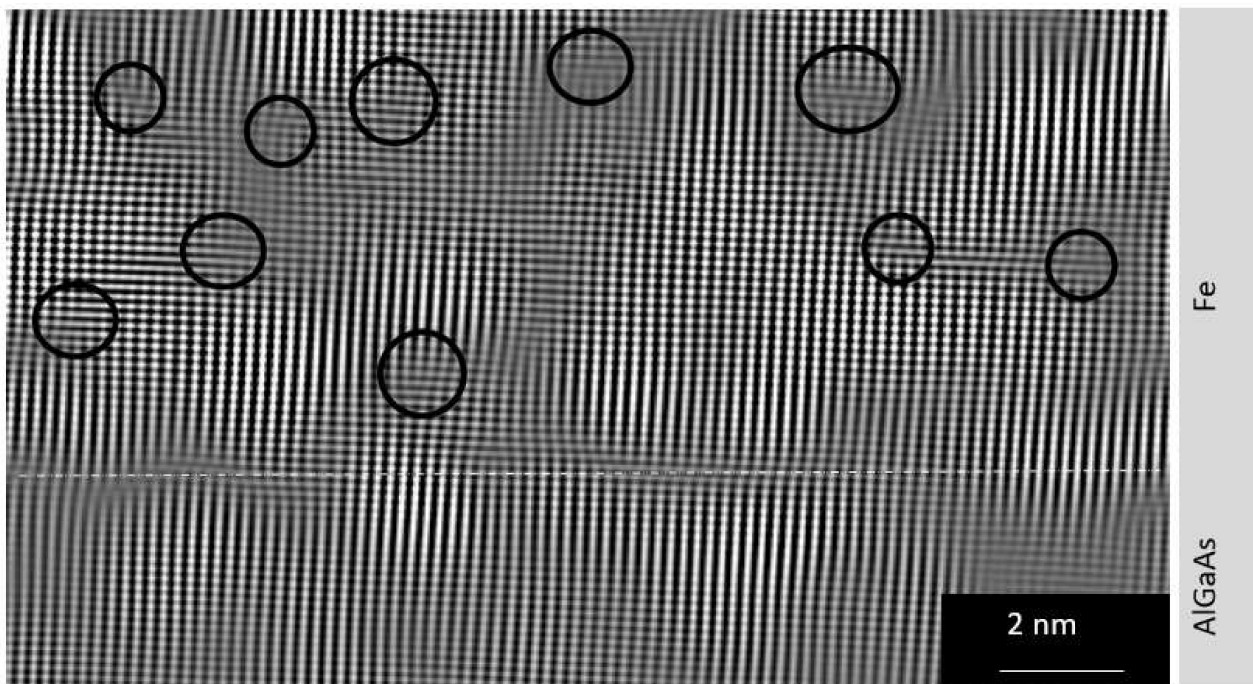

**Figure 6.** An IFFT image showing number of dislocations in Fe film deposited at room temperature.

It has been reported that the interfacial doping or intermixing, which determines the shape of the Schottky barrier and band bending, influences the spin injection efficiency [12]. Interdiffusion and reaction between Fe and GaAs have been demonstrated previously for certain growth conditions at higher temperatures. Note that the deposition temperature in the present case was lower, and the HRTEM studies at the interface showed no other reacted phase. However, the interdiffusion of Fe from the films to AlGaAs would occur at this deposition temperature, which could form an intermixing layer within GaAs. We employed fine-probe (probe size 0.5 nm) EDS mapping, as well as a line-scan for all the films, to study the intermixing behavior. Figure 7a is the HAADF image of the as-grown Fe film at 175 °C on an AlGaAs/GaAs/AlGaAs/GaAs (001) template, showing pronounced intermixing of Fe in the AlGaAs layer with an intermixing zone of around ≈15 nm. The corresponding composition line-scan showed a small shoulder, along with the primary Fe peak (see Figure 7b). However, such a diffuse region was not observed in the room-temperature-grown sample. The HAADF image and the corresponding composition line-scan of the room-temperature-grown Fe film on AlGaAs/GaAs/AlGaAs/GaAs(001) are shown in Figure 8a and Figure 8b, respectively. The intermixing zone in this case could not be detected.

Although no reaction product was observed at the Fe/GaAs interface, careful measurements showed different lattice spacing of (110) Fe upon annealing for room-temperature-deposited samples. From the HRTEM images, we estimated the lattice parameter of the room-temperature-grown sample and upon annealing at 200 °C for 1 h. The average (110) lattice spacing for the room-temperature-grown sample was ≈2.008 Å, while the average (110) spacing after annealing at 200 °C (see. Figure 9) was ≈2.022 Å. This suggested that the lattice parameter for the room temperature deposited sample was 2.840 Å, and after annealing it increased to 2.86 Å, which was close to the lattice parameter of bcc $\alpha$-Fe, 2.866 Å. The lattice spacing of Fe was calibrated with respect to the lattice spacing of the (111) GaAs.

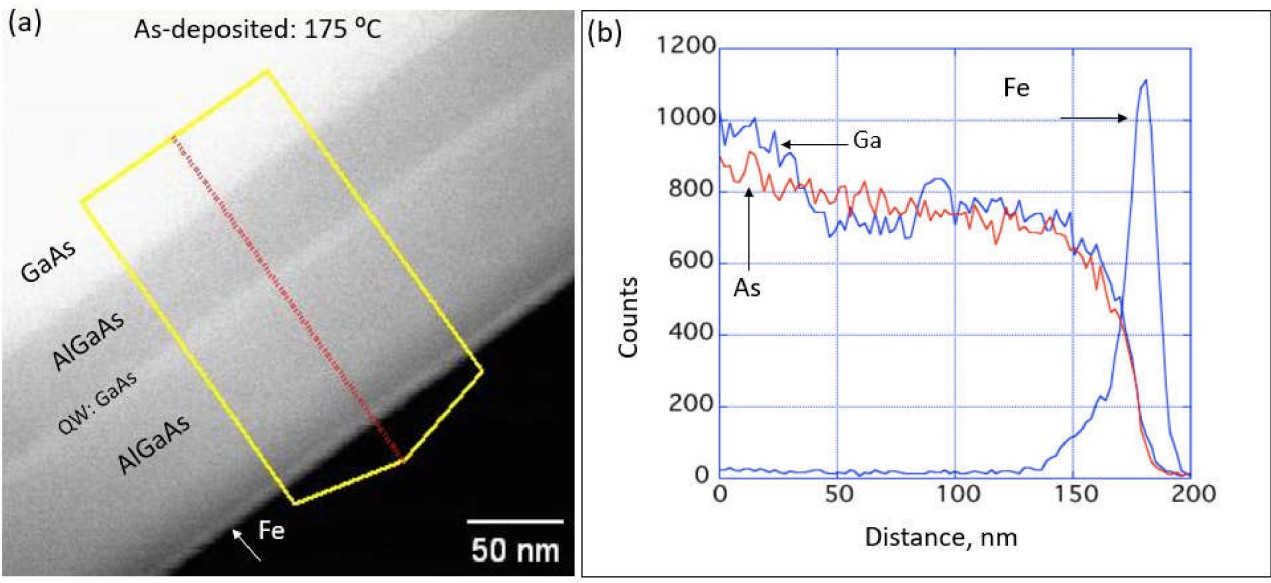

**Figure 7.** (**a**) An HAADF image of the Fe film deposited at 175 °C on AlGaAs/GaAs/AlGaAs/ GaAs(subs) template. (**b**) The corresponding compositional line-scan showing intermixing of Fe in AlGaAs at the interface.

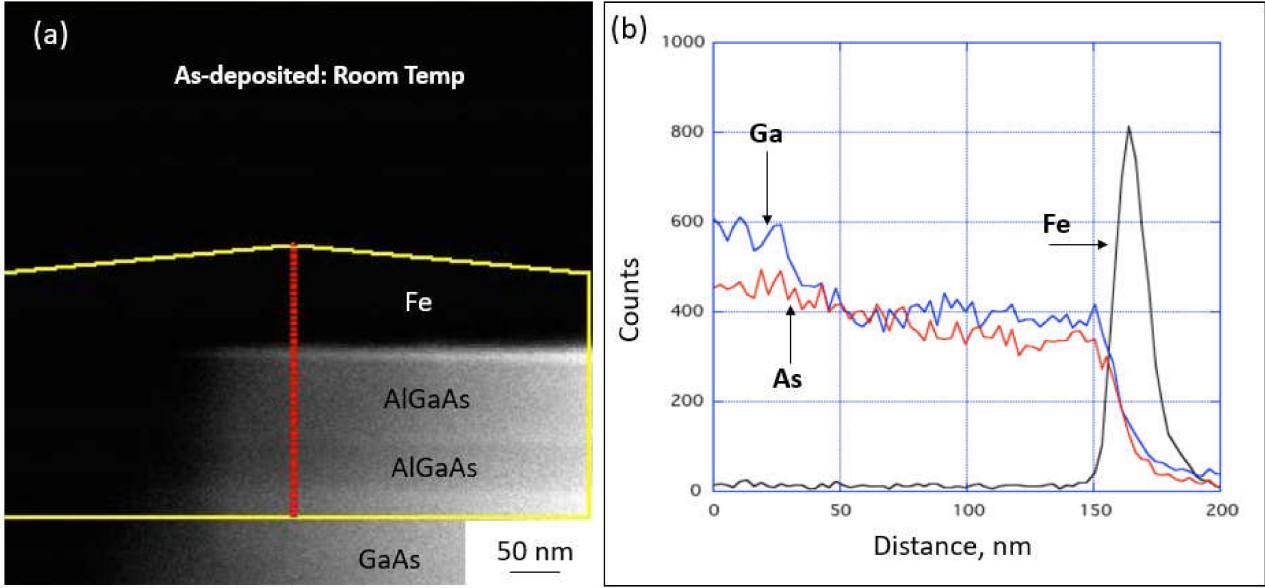

**Figure 8.** (**a**) An HAADF image of the Fe film deposited at room temperature on AlGaAs/GaAs/ AlGaAs/GaAs(subs) template. (**b**) The corresponding composition line-scan showing no intermixing.

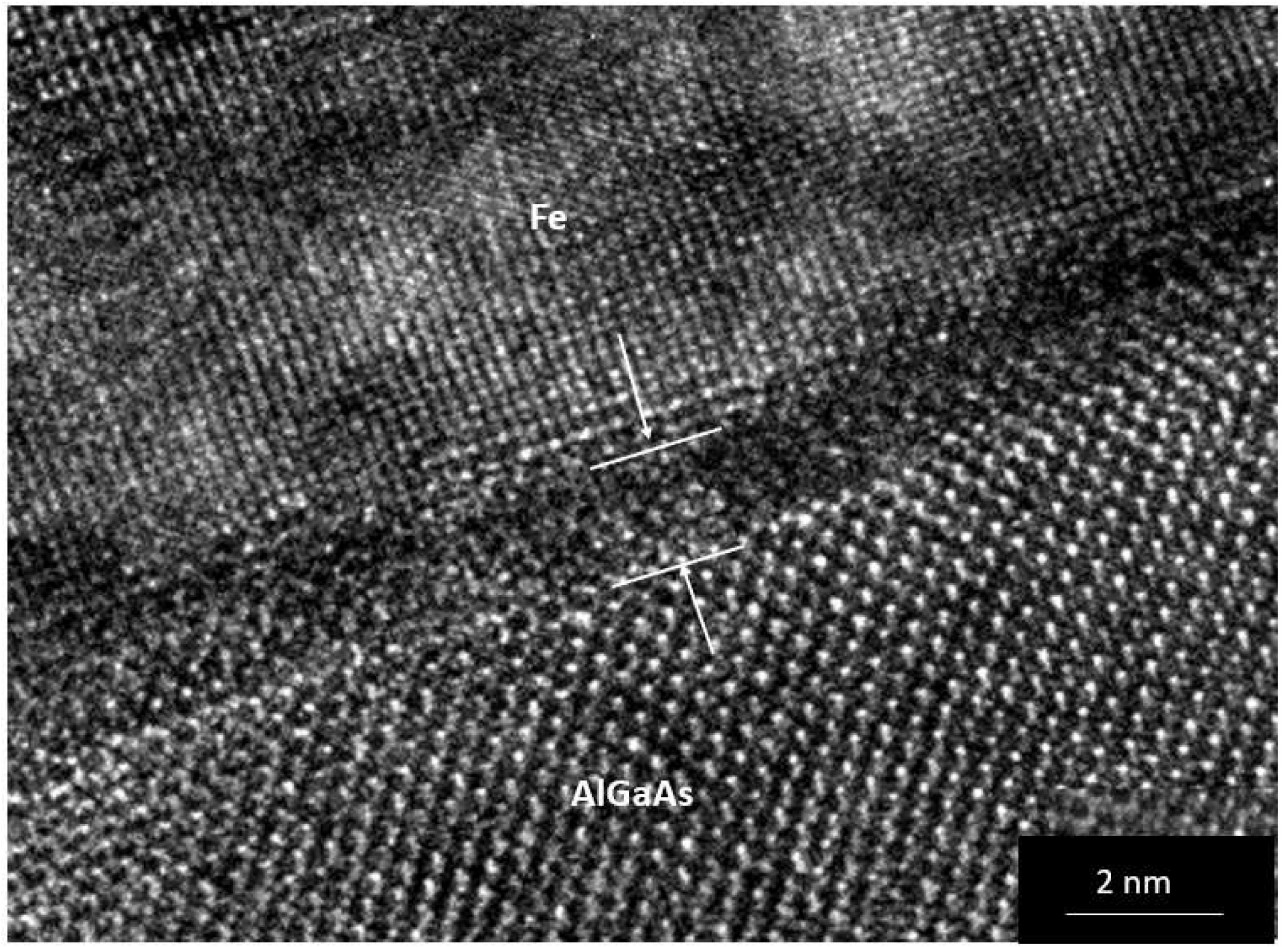

**Figure 9.** A HRTEM image showing the structure of the film and interfaces of the room-temperature-grown sample upon annealing at 200 °C. It can be observed that the interface is not sharp.

## 4. Conclusions

In summary, we investigated the microstructure, interfaces and phase formation in epitaxially grown Fe films of various thicknesses, 10 to 1000 nm, using molecular beam epitaxy at different temperatures on GaAs (011) and GaAs (001) substrates. High-resolution X-ray diffraction studies revealed that films with higher thicknesses were deposited as an equilibrium α-Fe, while the films of 10 nm thickness were mostly δ-Fe.

The measured lattice parameter of bcc-Fe on GaAs for the 10-nm-thick film was 2.835 Å, which was consistent with the δ-Fe with the lattice parameter, while the lattice parameter of the 1000-nm-thick film was 2.865 Å, consistent with the equilibrium α-Fe. Transmission electron microscopy revealed the interface for the 10-nm-thick films had strain lobes with no interfacial phase formation for films deposited at room temperature and 175 °C. However, at higher temperature the interface was not sharp, containing an intermixing layer in AlGaAs. We demonstrated that strain plays a major role in dictating the crystal structure of Fe film. The strain was significantly lower for the δ-Fe on GaAs compared to the α-Fe on GaAs. The intermixing behavior of Fe on AlGaAs was investigated using high-resolution transmission electron microscopy and HAADF imaging. At higher temperature, Fe diffused to the AlGaAs layer and formed an intermixing layer of (FeAl)GaAs.

**Author Contributions:** Conceptualization, Analysis and Writing: R.G.; Methodology and Analysis: S.B.Q. All authors have read and agreed to the published version of the manuscript.

**Funding:** Funding for this project was provided by the Office of Naval Research (ONR) through the Naval Research Laboratory's 6.1 Research Program.

**Institutional Review Board Statement:** Not applicable.

**Informed Consent Statement:** Not applicable.

**Data Availability Statement:** Data provided in the text as figures.

**Conflicts of Interest:** The authors declare no conflict of interest.

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
