# Peer review of "Evidence of Delta Phase of Fe in MBE-Grown Thin Epitaxial Films on GaAs"

_coatings, doi:10.3390/coatings12060771_

Round 1

Reviewer 1 Report

The authors present a systematic investigation of iron films grown on GaAs substrate grown by MBE by XRD and TEM. The results are based on the analysis of lattice parameters. There are some questions to be discussed. I recommend a careful revision as follows.

  • The “degree” symbols are mistyped in abstract and line 123;
  • What was the substrate temperature during growth? Some parts of the text are mentioned at room temperature; in others, 175 Celsius degrees, and others are still annealing procedure. Figure 4c denotes as grown at 175 oC, for example.
  • Do the authors employ elemental Fe? What about the cell temperature or atomic flux?
  • Do the authors perform RHEED during growth?
  • Are the XRD measurements of Fig. 2 performed in GID mode?
  • What about the roughness of films and their dependence on thicknesses?
  • I recommend that Fig 1 should be plotted between 43 and 47 degrees. The unity is missing in the abscissa axis.
  • Alpha and Delta phases of iron are very similar. Are there other manners to identify these phases?

Author Response

Reviewer-1

The authors present a systematic investigation of iron films grown on GaAs substrate grown by MBE by XRD and TEM. The results are based on the analysis of lattice parameters. There are some questions to be discussed. I recommend a careful revision as follows.

  • The “degree” symbols are mistyped in abstract and line 123;

Ans: Yes, we have corrected the typos in both places.

  • What was the substrate temperature during growth? Some parts of the text are mentioned at room temperature; in others, 175 Celsius degrees, and others are still annealing procedure. Figure 4c denotes as grown at 175 oC, for example.

Ans:

The films were grown at room temperature and at 175 C. The room temperature grown samples were then annealed at 200 C for one hour.  We have modified the text in the experimental section.

  • Do the authors employ elemental Fe? What about the cell temperature or atomic flux?

Yes, the target was elemental Fe. For more experimental details, we have cited the reference (see Ref. 11 to 14 ).

  • Do the authors perform RHEED during growth?

Ans: Yes, RHEED has been performed.

  • Are the XRD measurements of Fig. 2 performed in GID mode?

Ans:

GID mode was not required as the film is sufficiently thick. XRD in Fig. 2 was taken in specular or reflection mode.

  • What about the roughness of films and their dependence on thicknesses?

Ans:

We always find around 1 nm oxide layer formed epitaxilly on top of Fe, which might reduce the roughness. We have not measured the roughness as a function of thickness by AFM. It does not affect the outcome of the present findings.

  • I recommend that Fig 1 should be plotted between 43 and 47 degrees. The unity is missing in the abscissa axis.

Ans:

Thanks for your suggestions. We have let it as it covers the angular range 43 to 47 as suggested by the reviewer.  We incorporated the unit in the x-axis.  

  • Alpha and Delta phases of iron are very similar. Are there other manners to identify these phases?

Ans: High resolution XRD will be the efficient way to identify these phases.

Reviewer 2 Report

The authors proposed an experimental article with title "Evidence of Delta Phase of Fe in MBE Grown Thin Epitaxial Films on GaAs" based on molecular beam epitaxy on GaAs (011) and AlGaAs (001) substrates. The subject of the paper is interesting but there i believe that sufficient scientific report and theoretical background study is not suitable for this paper. after major revision with completely theoretical study and compare with experimental one, it can be reconsidered again. also, the references of the paper are poor and should be completed by authors. thus, in this state, my comment is major revision.

Author Response

Reviewer-2

The authors proposed an experimental article with title "Evidence of Delta Phase of Fe in MBE Grown Thin Epitaxial Films on GaAs" based on molecular beam epitaxy on GaAs (011) and AlGaAs (001) substrates. The subject of the paper is interesting but there i believe that sufficient scientific report and theoretical background study is not suitable for this paper. after major revision with completely theoretical study and compare with experimental one, it can be reconsidered again. also, the references of the paper are poor and should be completed by authors. thus, in this state, my comment is major revision

Ans:

This is an experimental paper on the evidence of delta-Fe phase at room temperature. We show experimentally the change in lattice parameter as a function of film thickness. Theoretical works suggested by the reviewer can be done for the future work. We have added seven additional references in the modified version.

Reviewer 3 Report

The author presented an interesting work to investigate the interfacial on the microstructure, interfaces and phase formation behavior in epitaxially grown Fe films. Actually, I am puzzled by IFFT-HRTEM image Figure 5 and Figure 6 that they seems to be of low quality and the readers cannot understand. Could they provide a better picture? Please!

I am also interested to see how the dislocations affect the optical properties and if they can measure conductivity of the samples it will be an important added value to the manuscript.

Author Response

Reviewer-3

The author presented an interesting work to investigate the interfacial on the microstructure, interfaces and phase formation behavior in epitaxially grown Fe films. Actually, I am puzzled by IFFT-HRTEM image Figure 5 and Figure 6 that they seems to be of low quality and the readers cannot understand. Could they provide a better picture? Please!

Ans: We have provided the greater quality pictures separately as tif images (300DPI).

I am also interested to see how the dislocations affect the optical properties and if they can measure conductivity of the samples it will be an important added value to the manuscript.

Ans: It’s a metallic film of Fe and the optical studies are beyond the scope of the present work. 

Reviewer 4 Report

The paper is well drafted and well organized. The proposed work is supported by experimental results and hence it is recommended to accept the after following minor revisions.

  1. The introduction is weak and can be enhanced by adding additional references.
  2. Add recent literature/ references and form a comparison table with the proposed work and their performance parameters with discussion.
  3. Replace all the figures with 300 dpi resolution.

Author Response

Reviewer-4

The paper is well drafted and well organized. The proposed work is supported by experimental results and hence it is recommended to accept the after following minor revisions.

1. The introduction is weak and can be enhanced by adding additional references.

Ans: We have added additional references and modified the introduction.

2. Add recent literature/ references and form a comparison table with the proposed work and their performance parameters with discussion.

Ans: As mentioned, we have added additional references to improve the introduction.

3. Replace all the figures with 300 dpi resolution.

Ans: All the Figures are sent as a separate TIF image file with 300 dpi resolution.

Round 2

Reviewer 1 Report

The authors corrected the points in question and the manuscript is ready for publication.

Reviewer 2 Report

Authors did not add theoretical study to their paper according my comment and i decided to reject it.

Reviewer 3 Report

-----